# Poly(Vinyl Alcohol)/Bovine Serum Albumin Hybrid Hydrogels with Tunable Mechanical Properties [note 1]

**DOI:** 10.3390/polym15234611

**Published:** 2023-12-04

**Authors:** Maria Bercea, Ioana-Alexandra Plugariu, Maria Valentina Dinu, Irina Mihaela Pelin, Alexandra Lupu, Adrian Bele, Vasile Robert Gradinaru

**Affiliations:** 1“Petru Poni” Institute of Macromolecular Chemistry, 41-A Grigore Ghica Voda Alley, 700487 Iasi, Romania; plugariu.ioana@icmpp.ro (I.-A.P.); vdinu@icmpp.ro (M.V.D.); impelin@icmpp.ro (I.M.P.); lupu.alexandra@icmpp.ro (A.L.); bele.adrian@icmpp.ro (A.B.); 2Faculty of Chemistry, Alexandru Ioan Cuza University of Iasi, 11 Carol I Bd., 700506 Iasi, Romania; robert.gradinaru@uaic.ro

**Keywords:** hybrid hydrogel, poly(vinyl alcohol), bovine serum albumin, mechanical behavior, elastic modulus, viscosity, self-healing ability, quasi-Fickian diffusion, bioadhesivity

## Abstract

In this study, a new strategy was adopted for obtaining polymer/protein hybrid hydrogels with shape stability and tunable mechanical or rheological characteristics by using non-toxic procedures. A chemical network was created using a poly(vinyl alcohol)(PVA)/bovine serum albumin (BSA) mixture in aqueous solution in the presence of genipin and reduced glutathione (GSH). Then, a second physical network was formed through PVA after applying freezing/thawing cycles. In addition, the protein macromolecules formed intermolecular disulfide bridges in the presence of GSH. In these conditions, multiple crosslinked networks were obtained, determining the strengthening and stiffening into relatively tough porous hydrogels with tunable viscoelasticity and a self-healing ability. A SEM analysis evidenced the formation of networks with interconnected pores of sizes between 20 μm and 50 μm. The mechanical or rheological investigations showed that the hydrogels’ strength and response in different conditions of deformation were influenced by the composition and crosslinking procedure. Thus, the dynamics of the hybrid hydrogels can be adjusted to mimic the viscoelastic properties of the native tissues. The dynamic water vapor-sorption ability, swelling behavior in an aqueous environment, and bioadhesive properties were also investigated and are discussed in this paper. The hybrid hydrogels with tunable viscoelasticity can be designed on request, and they are promising candidates for tissue engineering, bioinks, and wound dressing applications.

## 1. Introduction

Hydrogels are physically or chemically crosslinked networks with a structure and functionalities that are similar to living tissues, being suitable as artificial substitutes for damaged tissues [1,2,3]. Among other biological requirements (biodegradable, bioresorbable, no inflammatory response), they should provide support through tunable topography and stiffness for cell growth and proliferation with appropriate mechanical integrity [4,5].

Minimally invasive three-dimensional matrices are preferred to replace damaged tissues and, thus, biomolecule-based hydrogels are frequently tested for limiting undesirable effects [6,7]. However, the use of biomolecules is limited by their poor mechanical properties and accelerated degradation profile [8]. The modest tensile strength of many hydrogels restricts their use in load-bearing applications due to an early dissolution or the flow away from the local site [9].

Non-toxic synthetic polymers offer a suitable alternative for the design of networks with improved mechanical strength, but most of them present poor biocompatibility [10]. Synthetic hydrogels are promising materials in bioelectronics as soft supports to bridge rigid electronics and soft tissues [11], flexible strain sensors [12], versatile dynamic materials with a self-healing ability, and switchable mechanical properties or adhesive and shape-memory properties for tissue repair and replacement [13]. Interpenetrated networks with various shapes and pore sizes often represent a suitable way to design new materials with improved strength and stiffness that mimic biological tissues [14,15,16,17]. Porous interconnected network structures are major components of biological tissues. These networks can display various features, such as flexibility, strength or stiffness (monitored by the elastic modulus), toughness (measured by the fracture energy), or fatigue resistance (reflected by the fracture threshold). In order to meet all requirements, hybrid polymer/protein hydrogels are emerging as suitable porous support with a self-healing ability [18,19] that can be combined with cells and active biomolecules [20].

Poly(vinyl alcohol) (PVA) is a versatile nontoxic synthetic polymer extensively used for preparing new biomaterials with tunable mechanical properties that fall within the range of soft tissues [21,22,23]. The unique ability of PVA to induce excellent mechanical properties in physical or chemical networks stimulates research on the incorporation of this polymer into various composite hydrogels. Such approach considerably enlarges the application area of PVA-based materials, either as soft or tough networks, with the values of the tensile modulus in the range of 1 kPa to hundreds of MPa, as in natural tissues (skin, muscle, tendons) [21,24,25]. The strengthening and toughening of PVA hydrogels are efficiently adjusted via combined physical and chemical crosslinking [21]. However, biocompatibility and hemocompatibility investigations have shown that PVA hydrogels (obtained through a freezing/thawing procedure) used for vascular grafting are slightly irritant to the surrounding tissues [10].

Bovine serum albumin (BSA) was extensively investigated as a model globular protein, being one of the most important transporters for endogenous and exogenous molecules in the blood [26,27]. This globular macromolecule presents a predominantly α-helix structure (67%) [28] composed of around 583 amino acid residues. It contains three homologous domains (denoted I, II, and III with two subdomains for each domain: IA and IB, IIA and IIB, IIIA and IIIB), divided by 17 disulfide bonds into nine loops (L1–L9). Two binding sites of BSA, located in subdomains IIA and IIIA, were identified using molecular docking approaches [26]. At high temperatures, BSA macromolecules self-assemble into amyloidic structures due to a denaturation process when the α-helical structure is disrupted, determining a transition to β-sheets [29,30]. This change in the secondary structure of the protein is reversible below 50 °C but is only partially reversible above this temperature. At 60–65 °C, all the BSA helices are disrupted, and the conformational transition is irreversible.

In a dilute saline solution, at pH = 7.4 and a temperature of 37 °C, native BSA forms ideal mixtures with PVA (an “inert” macromolecular cosolute [31]), when the water-soluble polymer and the globular protein macromolecules exist as individual entities without intermolecular interpenetration. By increasing the total concentration of macromolecules, in a semidilute regime, a phase separation phenomenon occurs, and this increases the difficulty of using PVA/BSA mixtures for the design of stable composite materials with suitable mechanical properties.

Glutathione is the most abundant endogenous antioxidant tripeptide, which is present in mammalians cells (intracellular concentration between 1 and 15 mM [32]), with an important role in the defense against oxidative stress and reactive electrophile [33]. It is composed of glycine, cysteine, and glutamic acid, with a sulfhydryl group and a glutamyl moiety in its structure. In the healthy living cells, the ratio of reduced (GSH) to oxidized glutathione (GSSG) is higher than 10. Numerous cellular processes (cell proliferation and differentiation, immune response, signal transduction, cytokine production, etc.) are strongly controlled by the GSH/GSSG ratio [32,33].

GSH interacts spontaneously with serum albumins through hydrogen bonds and van der Waals forces [26,34], the process being enthalpy-driven [26]. BSA can be crosslinked through intermolecular protein disulfide bridges in the presence of GSH, forming an opaque gel at 37 °C and for pH values higher than 7. The slightly anionic form influences the BSA gelation, while the cationic form has no significant effect [34].

Genipin (GP) is a nontoxic crosslinking agent isolated from the fruits of the *Gardenia jasminoides*, with a high potential in the fabrication of stable networks for tissue engineering, wound healing, pharmaceutical, food and other biomedical applications [35,36]. GP is a selective crosslinking agent [37], being able to form crosslinks with the primary amine groups of chitosan, collagen, gelatin, or BSA (through the lysine units, BSA possessing a total of 59 lysine ε-amine groups with only 30–35 available groups) and to develop stable networks suitable for the tissue regeneration. The hydroxyl (–OH) or carboxyl (–COOH) functional groups are not targeted by genipin [37,38].

In this study, a new strategy was adopted to obtain hybrid hydrogels of PVA, BSA, and GSH with tunable viscoelastic properties. Multiple crosslinking approaches were used, i.e., dynamic S–S interactions, chemical crosslinking of BSA in the presence of genipin, and physical gelation of PVA through a freezing/thawing procedure. The main goal was to provide biomaterials with high stability, structural integrity, tunable viscoelastic properties, and rapid recovery after deformation as suitable systems for tissue repair and regeneration, bioinks, and wound dressing applications.

## 2. Materials and Methods

### 2.1. Materials

Poly(vinyl alcohol) (PVA), 99% hydrolyzed, with molecular weight (M) of 130 kg/mol; genipin (GP), purity of 98%, M = 226 g/mol; bovine serum albumin (BSA), M = 66.4 kg/mol; reduced glutathione (GSH), M = 307 g/mol; and glycerol, M = 92 g/mol were purchased from Sigma-Aldrich (Taufkirchen, Germany). Their chemical or 3D structures are given in Figure 1.

### 2.2. Hydrogel Preparation

The degree of the hydrolysis of PVA considerably influences its ability to form crystalline regions by applying freezing/thawing cycles. Therefore, a 99% hydrolyzed PVA sample with a high molecular weight was selected. The aqueous solution of PVA (concentration of 5 wt. %) was prepared through polymer dissolution in Millipore water at 90 °C under magnetic stirring until the polymer was completely dissolved (about 4 h); then, the solution was cooled at room temperature and filtered. A solution of 5 wt. % BSA was prepared by dissolving the protein in water at a low temperature (4 °C), and then GSH was added to the homogeneous solution (0.15 wt. % relative to protein) and weakly stirred. The PVA solution was mixed in different ratios with the BSA/GSH solution, and mixtures of various PVA/BSA compositions were prepared (Table 1). For ensuring a full hydration of protein, the pH value was adjusted to 7.4, when the negatively charged BSA adopts a more extended conformation [39,40].

The PVA/BSA/GSH solutions were incubated at 55 °C, when BSA macromolecules reached a partially disordered state [29], favoring S–S interactions between neighbored protein macromolecules [41] and also bridging between the protein and tripeptide [18]. The samples were then cooled down at room temperature. Genipin was added as a crosslinker (0.565 mmol/L or 1.13 mmol/L), and glycerol was added as a plasticizer (0.05 wt. %). The solubility of genipin in water is 1% (*w*/*v*) at 25 °C and 2% (*w*/*v*) at 37 °C [42]. Four samples (1, 4, 7, and 11) were not chemically crosslinked with genipin. All mixtures were poured into vials, stored at room temperature for 24 h, and then subjected to fast freezing in liquid nitrogen, followed by a thawing process at room temperature for about 24 h. The freezing/thawing cycles were repeated 3 times (samples 1–5, 7, 8) or 5 times (samples 6, 9–12), resulting in a variety of tough or soft hydrogels (Table 1). The samples were washed and immersed in Millipore water, being subsequently used for the rheological, mechanical, or bioadhesivity measurements.

Samples with similar compositions, prepared by using the procedure described above, were subjected to freeze-drying at −57 °C and 5.5 × 10^−4^ mbar for 72 h by using an ALPHA 1–2 LD Christ lyophilizer (Osterode am Harz, Germany) and then used for SEM, swelling and dynamic sorption investigations.

### 2.3. Scanning Electron Microscopy Investigation

The morphology of hydrogels was examined by using a Verios G4 UC Scanning Electron Microscope (SEM) from Thermo Scientific (Brno, Czech Republic), operating at 5 kV in a high-vacuum mode with a secondary electrons detector (Everhart–Thornley detector, ETD, Jeol, Tokyo, Japan). The average pores size was obtained as an average value of at least 60 randomly chosen pores from the micrographs by using Image J Software 1.8.0 for the morphology analysis according to the procedure described by Jayawardena et al. [43].

### 2.4. Mechanical Testing

Uniaxial compression tests were carried out at 25 °C on equilibrium-swollen hydrogels with a stable rod shape, diameter of about 9–10 mm, and height of 6–7 mm using a Shimadzu Testing Machine (EZ-LX/EZ-SX Series, Kyoto, Japan). All measurements were performed by applying a compression force of 450 N and a crosshead speed of 0.2 mm/min. Before each uniaxial compression test, an initial force of 0.02 N was set to be applied in order to ensure perfect contact between the hydrogel surface and the parallel compression plates of the mechanical analyzer. The strain (ε) was evaluated as the ratio between the change in length (Δ*l*) and initial length (*l_o_*). The compressive nominal stress (*σ*) was determined as the normal force (*F*) acting perpendicular to the rod area (*A*). Also, the compressive elastic modulus (*G*) was evaluated as the slope of the initial linear portion of the stress–strain curves according to a standard procedure previously reported for other porous materials [44,45]. All data were expressed as the means of two individual tests ± SD.

### 2.5. Rheological Measurements

The hydrogels with a softer structure were investigated under continuous and oscillatory shear conditions at 25 °C by using the modular compact rheometer Physica MCR 302 (Anton Paar, Graz, Austria) equipped with a plane-plane geometry (the diameter of the upper plate of 50 mm, gap of 500 μm) and a Peltier device. The viscoelastic behavior of the soft hydrogels was investigated in oscillatory shear conditions in the linear range of viscoelasticity determined for each sample in a preliminary amplitude sweep test. The elastic (G′) and viscous (G″) moduli were determined in frequency sweep tests for oscillation frequency (ω) between 0.1 rad/s and 100 rad/s at a strain (γ) value of 1%. G′ is a measure of the stored energy, and the G″ value reflects the amount of dissipated energy during one cycle of deformation. The loss tangent (tanδ = G″/G′) is correlated with the degree of viscoelasticity of the sample.

The shear viscosity (η) was measured in stationary shear conditions for the range of shear rates ($\dot{\gamma }$) from 0.01 s^−1^ to 100 s^−1^. A thixotropy test was carried out in a continuous shear mode for shear rates successively varied every 300 s from a low value, simulating the rest conditions (0.01 s^−1^), to high values (10 s^−1^, 50 s^−1^, 100 s^−1^, 500 s^−1^, or 1000 s^−1^) and again to the low step of the shear rate (0.01 s^−1^).

The creep and recovery behavior was investigated by applying a constant shear stress during the creep test (30 s), and then the shear stress was removed. The strain was monitored in time during the creep and recovery tests.

### 2.6. Dynamical Vapor Sorption Measurements

The behavior of hydrogels in the presence of moisture was investigated by monitoring the dynamic water vapor sorption ability using an automated gravimetric device IGAsorp made by Hiden Analytical (Warrington, UK) and equipped with ultrasensitive microbalance. The sample placed in a special container was dried at 25 °C in flowing nitrogen (250 mL/min) until its weight was constant, the relative humidity (RH) being less than 1%. Then, the RH was stepwise increased to 90% in steps of 10% RH with pre-established times (between 40 and 60 min) that ensure the equilibrium of sorption for each RH value.

### 2.7. Swelling Behavior

The swelling behavior of freeze-dried hydrogels was investigated at room temperature (25 °C) in water. The hydrogel was immersed in water, and it was taken out for weighing at different time intervals. The tests were repeated three times for each sample. The swelling degree (*S*), as the ratio between the solvent weight in the swollen state at a given time, *t* (*m_t_* − *m_o_*), and the weight of the corresponding dried sample (*m_o_*) are expressed as:(1)$$S=\frac{m_{t}-m_{o}}{m_{o}}\cdot 100\;(\%)$$

The disintegration of the networks was checked after 10 days of swelling in water at room temperature. The swollen samples were dried, and their weight was monitored until reaching a constant value (*m_dry_*). The mass loss for the hydrogel samples was determined as:(2)$$\mathrm{Mass}\mathrm{loss}=\frac{m_{o}-m_{dry}}{m_{o}}\cdot 100\;(\%)$$

### 2.8. Bioadhesivity

The bioadhesive performance of the hydrogels was investigated using a Brookfield Texture PRO CT3(R) texture analyzer (Brookfield Engineering Laboratories Inc., Middleboro, MA, USA). Fresh chicken skin mucosa was used as the model membrane. The membrane was horizontally attached to the lower end of the cylindrical probe (TA5; 10 mm diameter), and each sample was fixed to the TA-Ma fixture kit. The skin was placed in contact with the hydrogel sample, and a downward force of 1 N was applied for 60 s. Then, the skin was moved upward at a constant speed of 0.5 mm/s until attaining the complete separation of the surfaces. The adhesion force (the maximum value of the force required to detach the hydrogels from the membrane surface) was measured using TexturePro CT V1.9 Software. All measurements were performed at least three times.

## 3. Results and Discussion

The main advantage of PVA physical crosslinked networks obtained through the freezing/thawing procedure is the lack of toxicity. Thus, the porosity and mechanical properties of PVA hydrogels can be tuned during their preparation by selecting the appropriate concentration or molecular weight of the polymer and the number of applied freezing/thawing cycles [46,47]. However, the highly hydrophilic nature of PVA slows the cells’ proliferation and migration due to their poor adhesion and spreading [23,48]. By introducing bioactive macromolecules, such as adhesive proteins, into the PVA matrix, the cell compatibility was improved [48], and the hydrogels presented mechanical properties compared with those of porcine aorta [23]. The protein addition has no influence on the thermal behavior of PVA hydrogels; however, the physical treatments influence the crystallinity, as well as thermal and mechanical properties [49]. According to our observations, and as stated in the literature [23], PVA hydrogels obtained using the freezing/thawing method are stable up to approximately 70 °C. Above this temperature, the network structure is compromised due to the fact that PVA starts to dissolve in water.

Our goal was to design hybrid polymer/protein hydrogels that are stable in physiological conditions by using a water-soluble polymer, PVA, and a globular protein, BSA. Previously, it was shown that PVA can be considered an “inert” cosolute for BSA, and their mixture in a solution state presents an “ideal” behavior [31] when the intrinsic viscosity of a PVA/BSA mixture obeys the additive rule. The different types of macromolecules do not interact with each other; they only exist as individual entities surrounded by a solvent, without interpenetration. In these conditions, a phase separation appears for concentrated polymer/protein solutions at rest or under shear conditions. In order to prevent this phenomenon, in the first step, the solutions were heated at 55 °C to facilitate S–S interactions (the protein is slightly denatured) and cooled at room temperature when genipin and glycerol were added. Very fast freezing was performed in order to avoid the phase separation occurrence. The slow thawing step allows for PVA chains to generate crystalline regions. Thus, the PVA/PVA junction points and S–S dynamic bonds involving the protein and peptide ensure the stability of the network. The high protein content leads to weak hydrogels, which were investigated through rheology in various shear conditions. Supplementary chemical crosslinking was required to ensure the shape fidelity, and, thus, it was possible to investigate the sample in uniaxial compression tests. Due to its non-toxicity, genipin was selected as a crosslinking agent for preparing protein/polymer hybrid hydrogels, suitable either for bone regeneration or bioinks for skin tissue engineering [2,50]. The chemical crosslinking occurring between genipin and the primary amines groups present in BSA improved the shape stability of the hybrid network.

### 3.1. Morphology of Hydrogels

The micrographs provided though scanning electron microscopy (SEM) offer valuable information concerning the appearance of pores and the structural uniformity of the hydrogels [43]. Thus, the cross-sectional morphology of the PVA/BSA hybrid hydrogels was analyzed based on SEM images at different magnifications. The morphology of hydrogels is slightly influenced by the composition and applied crosslinking (Figure 1). The samples presented a network structure with interconnected pores formed by PVA chains through a physical procedure (by applying freezing–thawing cycles) or chemical crosslinking of macromolecules (in the presence of genipin for nine samples of different compositions, as shown in Table 1). In addition, dynamic disulfide bridges are formed between partially unfolded BSA macromolecules in the presence of GSH [18,26]. During the freezing–thawing process, water acts as a pore former. The pure solvent crystallizes first, and the ice crystals force the polymer chains to occupy the liquid part of the sample, forming regions of a high local polymer concentration. This state with PVA chains close to each other favors the strong –OH intermolecular interactions and the formation of crystalline zones during the slow thawing process [46,47] After applying several successive freezing/thawing cycles, the overall system presents a heterogeneous structure with phase-separated domains, i.e., polymer-rich regions and areas predominantly occupied by the solvent [46]. The crystalline regions with PVA-PVA junction points act as knots in a three-dimensional network [18,46,47]. The cleavage and rearrangement of disulfide bonds occurring in BSA are triggered by the presence of GSH [26,34] and a gentle thermal treatment before applying the freezing/thawing cycles, resulting in a soft disulfide bridge network. The additional chemical crosslinking in the presence of genipin ensures the development of stable multiple networks as a result of the multiple interactions that are established between the components of these systems.

The hybrid hydrogels present a discontinuous structure with interconnected pores (Figure 1). It can be observed that by increasing the PVA content in the 3D network, the average pore size decreases from 40–50 μm for samples 7–9 to 20–30 μm for samples 1–6. The pore size homogeneity decreases considerably for samples 8, 9, and 12, whereas sample 11 presents a high frequency of deep holes (>50 μm). When BSA is predominant in the hydrogel and genipin content increases (sample 10, Figure 1k), clusters and aggregates are formed, the porous structure is perturbed, and BSA-rich zones appear in the network. In the absence of PVA, the network formed by crosslinking BSA with genipin and by S–S interactions (sample 13, Figure 1n) presents a more homogeneous wall structure and pore distribution, as compared with sample 10.

### 3.2. Mechanical Properties

Figure 2 presents the mechanical behavior of hybrid hydrogels of different compositions in uniaxial compression tests, when a nominal force of 450 N was applied. A typical elastic behavior, which is commonly observed in porous materials, is depicted for the PVA/BSA-based hydrogels. All tested hydrogels demonstrated a high degree of resilience; they were able to sustain compression values beyond 80% without any crack development or failure of the gel network (Figure 2a). This indicates a remarkable level of flexibility and structural integrity, even under significant compressive loads.

The introduction of BSA (up to 30%) in the PVA matrix and its crosslinking with genipin led to an increase in compressive strength, as illustrated in Figure 2b. This increase is indicative of a rise in network rigidity, a change that can be explained by a dual crosslinking mechanism: (1) a physical crosslinking through hydrogen bonds due to the presence of PVA and (2) a chemical crosslinking through the action of genipin, which creates a strong, irreversible network within the material. Due to this dual crosslinking effect, the hybrid hydrogels (PVA/BSA, chemically crosslinked with genipin) exhibited an enhanced compressive strength compared to the physically crosslinked PVA hydrogel. For example, the PVA/BSA hydrogels (samples 2 and 3) were able to sustain high compressions of 87.18% and 92.67%, respectively, at compressive nominal stresses of 1.788 MPa and 1.999 MPa. In comparison, the PVA hydrogel (sample 1) sustained 88.41% compression at a compressive nominal stress of 1.62 MPa (as shown in Figure 2b).

The compressive moduli of PVA and PVA/BSA hydrogels, calculated as the slope of the initial linear portion in the σ–ε profiles (Figure 2c), were modulated by varying the ratio between PVA and BSA or through the addition of a higher amount of genipin. For instance, sample 3 containing 30% BSA exhibited a compressive elastic modulus of 13.66 kPa and a recovery degree of 28.42%, while sample 5 containing 50% BSA showed an elastic modulus about 3.5 times higher and a recovery degree of 54.54%. An increase in the genipin and BSA content (sample 10) determines a value of the elastic modulus of 14.33 kPa, being in the same range as those obtained for sample 3 but with a higher degree of recovery (43.77%—sample 10, compared to 28.42%—sample 3, Figure 2d). The compressive elastic modulus values obtained for sample 3 and sample 10 indicate high elasticity of the networks, which can be associated with the less dense internal morphology of the hydrogels. In these samples, small pores are evident in the walls of the macropores (see SEM images, Figure 1c,k). These results are in agreement with the data reported for fibronectin-functionalized physically crosslinked poly(vinyl alcohol) [23], PVA/Phytagel [22], chitosan/agarose [51], and chitosan/dextrin hydrogels [52].

The results of this study suggest that the compressive modulus values were influenced by the PVA/BSA ratios and the chemical crosslinking with genipin. Additionally, the recovery degree also varied, indicating differences in the materials’ ability to return to their original shape after deformation. It appears that, for sample 6, a high degree of recovery (82.12%) was achieved. This improvement was attributed to a specific treatment involving a 1:1 weight ratio of PVA to BSA and a higher number of freezing/thawing cycles. The mechanical enhancement resulting from an increased number of freezing/thawing cycles is explained by the involvement of PVA crystallites [18,46,47]. These crystallites serve as physical crosslinks within the hydrogel structure. In this context, the system can be conceptualized as a network where these crystallites are trapped within a polymeric matrix formed during the cryogelation process. The crystallites essentially act as knots within the network, ensuring significant dimensional stability and elastic properties of the hydrogels [47,52,53]. The values of the compressive elastic modulus between 14 kPa and 50 kPa make these hydrogels suitable for low-load-bearing tissue repair and regeneration [1,17,54,55,56].

### 3.3. Rheology

Dynamic oscillatory measurements allow for the evaluation of the material responses in real time. Figure 3 presents the dependence of the elastic moduli (G′) and loss tangent (tanδ) on the oscillation frequency (ω) for the soft hydrogel samples, which could not be mechanically tested through compression tests. It can be observed that the viscoelastic properties of sample 4 (obtained by applying freezing–thawing cycles) and sample 9 (physical and chemical crosslinked networks) are very close, and the value of G′ exceeds, by more than one order of magnitude, those of sample 11, whereas the loss tangent is less than half. Also, the strength of physical crosslinked networks is dependent on the PVA content: samples 7 and 11 are weak hydrogels, as compared with sample 4 (the same amount of polymer and protein) or sample 1 (pure PVA hydrogel). In the absence of PVA, BSA forms a weak network with a stable structure (sample 13).

The flow behavior also depends on hydrogel composition and crosslinking procedure. Figure 4 shows the dependence of the apparent viscosity (*η*) on shear rate, $\dot{\gamma }$ (Figure 4a), and shear stress, *τ* (Figure 4b). Above 0.1 s^−1^, *η* decreases with increasing shear rate and scales as *η*~${\dot{\gamma }}^{-n}$, where the flow index, *n*, is between 0.68 and 0.76 for the hybrid PVA/BSA hydrogels. The resistance to flow, reflected by the viscosity values, decreases from sample 4 and 9 (stronger networks) to sample 8 (intermediate network strength) and samples 7, 12, and 11 (weaker networks). The lowest viscosity value was registered for a pure BSA hydrogel (sample 13), which is close to that reported in the literature for a BSA hydrogel obtained from a 5% solution through a chemical crosslinking with epichlorohydrin [57]. The yield stress (τ_o_) represents the minimum shear stress, which is required to initiate the flow (Figure 4b). The value of τ_o_ varies from 40 Pa (sample 11) to about 1000 Pa (sample 4) and can be correlated with the mechanical strength of the hybrid hydrogel. A pure protein hydrogel (sample 13) has a very low τ_o_ value (0.82 Pa) as compared with PVA/BSA hydrogels, showing good moldability (Figure 4b).

For the hydrogels, an important requirement concerns the thixotropic behavior. From a structural point of view, it is important that the intermolecular interactions that ensure the structural integrity are, to a large extent, quickly recovered after deformation. For the hybrid hydrogels with a strong-enough network at rest, the viscosity decreases when shear forces are applied, and it recovers very quickly to the initial value after removing the shear forces. Figure 5 presents the self-healing behavior of sample 9 subjected to a cycle of three steps of shear rates: low (0.01 s^−1^, in the Newtonian region, Figure 4)—high (from 10 s^−1^ to 1000 s^−1^)—low (0.01 s^−1^) values. The hydrogel presents a self-healing ability, recovering its structure completely for $\dot{\gamma }$ ≤ 100 s^−1^. For higher shear rate values, the structure is affected by the shear forces, and the initial state of equilibrium is reached in a longer time or only partially recovered if the $\dot{\gamma }$ value during step 2 was very high.

The hydrogel samples were subjected to cycles of increasing shear stress values during the creep test (Figure 6), which indicates a transient response, including a high value of an instantaneous elastic strain (γ_iel_), a much smaller value of the delayed elastic strain (γ_del_), and a permanently viscous part (γ_vis_). When the action of shear stress is removed, γ_iel_ quickly recovers, γ_del_ recovers in time, and γ_vis_ at equilibrium represents the irreversible deformation (Figure 6b). For τ < 300 Pa, the γ_iel_ values are very high, and γ_vis_ = 0, suggesting that the hydrogel completely recovers its structure after deformation (Figure 6a,c). It can be seen that the recovery time increases with an increasing τ value. An irreversible deformation of the network was registered for weak network structures (Figure 6b); when a high shear stress was applied (Figure 6d), the structure was only partially recovered. The recovery degree decreased above 1000 Pa, and for τ > 1200, the viscous deformation (irreversible) was above 90%.

The physical network formed by PVA after applying freezing/thawing cycles contributes to the increase in the hydrogel elasticity. In addition, the chemical network formed by BSA crosslinked with genipin ensures the resistance to strain during repetitive cycles of deformation. The S–S interactions developed in the hybrid hydrogels contribute to the self-healing ability [18,26] and the recovery degree after compression (Figure 2) or creep (Figure 6) tests. Thus, the dynamics of the hybrid hydrogels can be adjusted to mimic the viscoelastic properties of the native tissues that exhibit a stress relaxation behavior [55].

### 3.4. Dynamic Vapor Sorption

The hydrophilicity of hydrogels is a crucial characteristic that provides a desirable property to enhance cellular activities, and it influences the moisture of the hydrogels. A moist environment promotes wound healing by enabling the encapsulated cells to migrate freely through the hydrogels in contact with the tissue. On the other hand, mechanical and bioadhesive properties get worse if the amount of water or biological fluids is too high; thus, a compromise is necessary to induce optimal properties to the new hydrogel. The following steps were identified during the adsorption of water vapor through the porous hydrogels: (a) the adsorption of water molecules on the outer surface of the hydrogel, (b) transfer of water molecules to the internal pores’ space, and (c) the final diffusion of water molecules into the pores through capillary condensation [58].

The dynamic water vapor sorption ability of hybrid hydrogels was monitored for a selection of hydrogels (samples 3, 5, 6, and 9, Table 1). Figure 7 shows the weight changes (*W*) as a function of relative humidity (*RH*), evidencing a hysteresis in the water vapor adsorption/desorption isotherms. The progressive adsorption of water vapor from 0 to 10% d.b. is registered for all samples when *RH* increases up to 50% (almost linear plots in Figure 7). For *RH* > 50%, a sharp increase in water sorption occurs, reaching the maximum values for *RH* values around 90%.

The hysteresis loops are generally related to capillary condensation in open-ended pores (as for sample pores of a cylindrical geometry) [59]. Due to the capillary forces, the desorption of water molecules from the pores is slower than their adsorption (the amount of water vapor is less during desorption than during adsorption).

The Brunauer–Emmett–Teller (BET) kinetic model was applied to experimental sorption data registered under dynamic conditions, and the specific surface area was evaluated (Table 2):(3)$$W=\frac{C\;\times W_{m}\;\times \;RH}{\left(1-RH\right)\;\left(1-RH+C\;\times RH\right)}$$
where *W*—the weight change (the amount of adsorbed water); *W_m_*—the weight of water forming a monolayer; *RH*—the relative humidity; *C*—an adsorption constant.

The BET model describes the adsorption isotherm types I–IV for relative humidity up to a value of about 40%. In our study, the highest BET values were obtained for the physically and chemically crosslinked hydrogel with a BSA content higher than that of PVA.

### 3.5. Swelling Kinetics

The swelling investigations carried out in water at room temperature evidence a fast swelling for all samples, the maximum swelling degree being reached in no more than 10 min (Figure 8a). For a high BSA content, a faster initial swelling rate after the hydrogel’s immersion is obtained. The value of the swelling degree at equilibrium (*S*_eq_) depends on the hydrogel composition and network stability in water for a longer period of time (Figure 8b,Table 1).

The water diffusion through the hydrogel pores was analyzed by using the following equation:(4)$$F(t)=\frac{m_{t}}{m_{\infty }}=k\;t^{n_{s}}$$
where *F*(*t*) is the total water uptake into the hydrogel sample at a given time, *t*; *m_t_* and $m_{\infty }$ are the quantity of solvent absorbed at a given time, *t*, and the quantity of solvent absorbed by the hydrogel at the equilibrium of swelling, respectively; *k* represents a characteristic rate constant which is influenced by the network structure; *n_s_* is the transport number correlated with the diffusion and/or relaxation phenomena that control the swelling process.

Equation (4) was applied to the early swelling stage. From the slope of linear dependences of *F*(*t*) versus time (*t*), the diffusion exponent values, *n_s_*, were obtained (Table 1). These values reveal the solvent transport mode through the pores of hybrid hydrogels. The swelling data results obtained for all samples led to *n_s_* values lower than 0.5, suggesting a quasi-Fickian diffusion of water through the hydrogels.

At a high BSA content, the swelling degree of the hydrogel increases, but the mass loss also becomes important. This is the result of the network disintegrating during immersion in water (Table 1). For a high BSA content, the samples present clusters and aggregates of protein trapped within the weak PVA network (Figure 1k). These aggregates harmed the hydrogels’ integrity, determining weak disintegration in an aqueous environment. After 10 days of immersion in water, a slight weight loss was observed for samples 1 to 10 (below 3%), and the weight loss increased for samples 11–13 (8.26% for sample 11, 6.11 for sample 12, and 10.18% for sample 13). For the BSA hydrogel crosslinked with epichlorohydrin, a degradation of 12% was observed after 3 days [57].

### 3.6. Bioadhesivity Evaluation

The PVA/BSA hybrid hydrogels present elasticity and are easily deformable; thus, they can adapt to the shape of the surface to which they are applied. In this context, the bioadhesive properties can be helpful in their immobilization at the site of targeted application or for adhesion on different surfaces that are not horizontal [9]. Therefore, protein-based hydrogels can act as tissue adhesives with a long contact time, providing artificial cellular microenvironments to stimulate tissue regeneration or facilitate drug administration [60]. For biomedical and pharmaceutical applications, the bioadhesive property is in correlation with the elastic properties and self-healing ability of hydrogels [61,62]. The structure of the hybrid materials can be tuned to provide optimal rheological properties and mechanical resistance during the attachment and removal from the site of application.

Bioadhesive tests were carried out for PVA/BSA hydrogels in contact with a fresh chicken skin surface, and the adhesion force was evaluated. The values of the adhesion force obtained for most hydrogel samples were above 0.3 N (Figure 9) and fell within the required range of bioadhesion, between 0.3 N and 1.4 N (*p* < 0.05) [62,63]. The highest values of the adhesion force were obtained for samples 5 and 6 (1.38 N and 1.06 N, respectively).

Samples 4 and 9 presented similar bioadhesivity, with an adhesion force of 0.63 N–0.64 N. These BSA-based hydrogels can play the role of emergent bioadhesives [64]; they can cling to biological surfaces without the need for a glue. This feature can be attributed to the formation of adhesion junctions at the hydrated interface due to physical or dynamic covalent interactions between the different surfaces in contact. Dynamic S–S bonds established between protein macromolecules or between BSA and GSH, as well as hydrogen bonds between the available functional groups (hydroxyl or primary amine groups), increase the bioadhesion of hybrid protein/polymer hydrogels.

Bioadhesivity is an important characteristic of hydrogels, being influenced by the structure and density of the network. Its occurrence improves the compatibility of materials with living organisms. The adhesive properties are poor if there is an excessive absorption of water from biological fluids into the pores (samples 11 and 12), especially at the hydrogel surface, decreasing the interfacial interactions between the hydrogel and the targeted tissue [65,66].

An efficient design of hybrid hydrogels must take into account the possibility for tailoring the viscoelastic response in correlation with the adhesion strength, functionality, and stability in physiological conditions. The incorporation of BSA and tripeptide, which is able to form dynamic S–S interactions, induces superior adhesive properties to PVA hydrogels, improving their compatibility with the living organisms. However, an optimum crosslinking degree must be chosen for conserving the required porosity and water adsorption, maintaining the targeted mechanical/rheological properties of hybrid hydrogels.

Bioadhesivity and mechanical properties provide the required physical strength and support for load-bearing applications. The PVA network obtained through a freezing/thawing procedure has no cytotoxic effects [67]. According to Figure 2 and Figure 9, the PVA hydrogel presents good mechanical properties and an acceptable value of adhesion force. By incorporating BSA, which is naturally sticky [68], the hybrid hydrogels present improved adhesion because the protein is able to establish interactions with the biological substrate.

## 4. Conclusions

By combining the extraordinary mechanical properties of PVA networks with the biological characteristics (bioadhesiveness, biocompatibility) of BSA, multiple protein/polymer networks with a self-healing ability were designed. Tailorable properties can be induced to the hybrid hydrogels through multiple crosslinking strategies using combined physical and chemical methods. The viscoelasticity can be tuned by selecting an adequate composition of the polymer and protein and a suitable crosslinking procedure specific for each system. Thus, freshly prepared and well-homogenized PVA/BSA aqueous solutions in the presence of GSH were heated to allow for the formation of S–S bonds between BSA macromolecules in a partially disordered state. Furthermore, a natural crosslinking agent (genipin) was added for the chemical crosslinking of BSA at room temperature, and then several freezing–thawing cycles were applied for the physical crosslinking of PVA. A SEM analysis showed the formation of porous interconnected three-dimensional networks, which serve as permeable biomaterials for the transport of biological fluids and bioactive compounds.

The physically and chemically double crosslinked hydrogels presented a compact structure and enhanced mechanical properties. The rheological and mechanical behavior of these hydrogels can be tailored and controlled through the number of freezing/thawing cycles, the content of the chemical crosslinker, and a careful selection of the ratio between the macromolecules, PVA, and BSA. The most promising materials for soft tissues were obtained using combined methods of crosslinking applied to solutions with a high PVA content (physical through freezing/thawing or chemical through a crosslinking agent and reactive functional groups) or by varying the concentration of the crosslinking agent. However, the weak hydrogels (obtained for a low polymer content or in the absence of PVA) can be of interest as injectable materials or wound dressing applications.

The BSA-containing networks presented non-Fickian diffusion of water molecules through the pores. Also, they showed improved bioadhesivity, this characteristic being required for the adhesion of hydrogels on biological tissues.

The main advantage of the composite polymer/protein system is that the network properties can be modulated during their preparation through optimum physical/chemical crosslinking, generating a variety of viscoelastic features from relatively tough to weak hydrogels. Thus, the elastic modulus of the hybrid hydrogels can settle within suitable ranges to mimic the biological soft tissues. The multiple crosslinked PVA/BSA networks provide suitable viscoelasticity, shape integrity, excellent self-healing ability, and improved bioadhesion, properties that are recommend for these hybrid materials as potential candidates for tissue repair and regeneration, bioinks, and wound dressing applications.

## Data Availability

Data are available on request.

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
