# Peer review of "Poly(Vinyl Alcohol)/Bovine Serum Albumin Hybrid Hydrogels with Tunable Mechanical Properties†"

_polymers, 2023, doi:10.3390/polym15234611_

Round 1

Reviewer 1 Report

Comments and Suggestions for Authors

This manuscript reported a PVA/BSA-based hydrogel with dynamic and switchable features including mechanical and self-healing behaviours via the construction of double network between PVA and BSA. This work is interesting and novel, which can be publication in polymers when solving the following issues.

1.      For the Introduction Part, some contents can be combined into a single paragraph, so that it reads more smoothly.

2.      The comment in line 105-108 is too little, the authors should focus on and describe the unique structures, properties, and application potential for the developed hydrogels.

3.      The authors should add the optical, digital images of dynamic hydrogel between soft and stiff states and test its mechanical tensile stress-strain behaviours in the revised manuscript.

4.      In addition, this hydrogel should be demonstrated with good flexibility and foldability.

5.      Although the hydrogel shows good switchability in mechanical and self-healing performance, the authors need to further explore and analyze its internal mechanism, which is very important for designing functional gels. These works (Advanced Materials, 2022, 34(10): 2107857; Polymers, 2022, 14(24): 5338;) should be discussed and referenced in the revised manuscript to illustrate the dynamic mechanism of the PVA/BSA-based hydrogel.

6.      Compared to other hydrogel, the advantages of this PVA/BSA-based hydrogel should be discussed, and its potential application, such as bioelectronics, tissue repair, or wound dressing should also be investigated in the revised manuscript.

7.      For the Conclusions Part, the authors should refine the content, highlighting the advantages of the PVA/BSA-based dynamic hydrogel.

Comments on the Quality of English Language

A lot of useless statements.

Author Response

This manuscript reported a PVA/BSA-based hydrogel with dynamic and switchable features including mechanical and self-healing behaviours via the construction of double network between PVA and BSA. This work is interesting and novel, which can be publication in polymers when solving the following issues.

  1. For the Introduction Part, some contents can be combined into a single paragraph, so that it reads more smoothly.

Introduction was revised and improved. Some paragraphs and references were removed.

  1. The comment in line 105-108 is too little, the authors should focus on and describe the unique structures, properties, and application potential for the developed hydrogels.

The sentence was improved:

In this paper, a new strategy was adopted to obtain hybrid hydrogels of PVA, BSA and GSH with tunable viscoelastic properties. Multiple crosslinking approaches were used, i.e., dynamic S–S interactions, chemical crosslinking of BSA in the presence of genipin and physical gelation of PVA by freezing/thawing procedure. The mail goal was to provide biomaterials with high stability, structural integrity, tunable viscoelastic properties and rapid recovery after deformation, as suitable systems for tissue repair and regeneration, bioinks or wound dressing applications.

  1. The authors should add the optical, digital images of dynamic hydrogel between soft and stiff states and test its mechanical tensile stress-strain behaviours in the revised manuscript.

Digital images of a dynamic PVA/BSA hydrogel were added. Other mechanical tests and the thermal behavior will be presented elsewhere.

  1. In addition, this hydrogel should be demonstrated with good flexibility and foldability.

We thank the reviewer for this comment. We will take intro account for a further investigation. For samples investigated in the present study, the flexibility increases by adding BSA. Foldable strucures can be obtained from weak hydrogels. The strong network structures investigated by mechanical tests recover very fast their initial shape and they are not foldable.

  1. Although the hydrogel shows good switchability in mechanical and self-healing performance, the authors need to further explore and analyze its internal mechanism, which is very important for designing functional gels. These works (Advanced Materials, 2022, 34(10): 2107857; Polymers, 2022, 14(24): 5338;) should be discussed and referenced in the revised manuscript to illustrate the dynamic mechanism of the PVA/BSA-based hydrogel.

The discussion concerning the PVA/BSA hydrogel formation was improved. Among other articles of interest, the indicated references were read and included in the revised manuscript:

- Zhao, D.; Pang, B.; Zhu, Y.; Cheng, W.; Cao,  K.; Ye, D.; Si, C.; Xu, G.; Chen, C.; Yu, H. A stiffness-switchable, biomimetic smart material enabled by supramolecular reconfiguration. Adv. Mat. 2022, 34(10), 2107857. https://doi.org/10.1002/adma.202107857.

- Moxon, S.R.; Richards, D.; Dobre, O.; Wong, L.S.; Swift, J.; Richardson, S.M. Regulation of mesenchymal stem cell morphology using hydrogel substrates with tunable topography and photoswitchable stiffness. Polymers 2022, 14, 5338. https://doi.org/10.3390/polym14245338

  1. Compared to other hydrogel, the advantages of this PVA/BSA-based hydrogel should be discussed, and its potential application, such as bioelectronics, tissue repair, or wound dressing should also be investigated in the revised manuscript.

Our interest was to obtain hydrogels suitable for tissue repair or wound dressing applications. We tested the hydrogels only in aqueous environment.

For bioelectronics, we will investigate the hybrid hydrogels in salted environment in a future study.

<<  Bioadhesivity and mechanical properties provide the required physical strength and support for load-bearing applications. PVA network obtained by freezing/thawing pro-cedure has no cytotoxic effects [67]. According to Figures 2 and 9, PVA hydrogel presents good mechanical properties and an acceptable value of the adhesion force. By incorpo-rating BSA, which is naturally sticky [68], the hybrid hydrogels present improved adhe-sion because the protein is able to establish interactions with the biological substrate.>>

<<  The main advantage of the composite polymer/protein system is that the network properties can be modulated during their preparation by optimum physical/chemical crosslinking, generating a variety of viscoelastic features, from relative tough to weak hydrogels. Thus, the elastic modulus of the hybrid hydrogels can be settled within suitable ranges to mimic the biological soft tissues. The multiple crosslinked PVA/BSA networks provide suitable viscoelasticity, shape integrity and excellent self-healing ability, also improved bioadhesion, properties that recommend these hybrid materials as potential candidates for tissue repair and regeneration, bioinks or wound dressing applications.>>

  1. For the Conclusions Part, the authors should refine the content, highlighting the advantages of the PVA/BSA-based dynamic hydrogel.

Section Conclusions was revised and improved.

Comments on the Quality of English Language:

A lot of useless statements.

The manuscript was carefully revised and many sentences were reformulated.

We thank the reviewer for the analysis of the manuscript and for the useful comments and suggestions.

Reviewer 2 Report

Comments and Suggestions for Authors

The manuscript entitled "PVA/BSA hybrid hydrogels with tunable viscoelastic properties"  deals with the fabrication of hybrid hydrogel using non-toxic materials. The idea is presented clearly and a number of experiments are carried out related to viscoelastic properties. However, the major drawback of this manuscript is the lack of statistical analysis.

1. Line 79-80: Please explain the meaning of the "ideal" mixtures.

2. Authors are recommended to perform statistical analysis to prove the significance and reproducibility of the obtained results. 

3.. Section 4 is missing in this manuscript. I assume that the authors forgot to include a discussion section in the manuscript. A detailed discussion section comparing existing literature and insights must be included in the manuscript.

Author Response

The manuscript entitled "PVA/BSA hybrid hydrogels with tunable viscoelastic properties" deals with the fabrication of hybrid hydrogel using non-toxic materials. The idea is presented clearly and a number of experiments are carried out related to viscoelastic properties. However, the major drawback of this manuscript is the lack of statistical analysis.

  1. Line 79-80: Please explain the meaning of the "ideal" mixtures.

The comments concerning the ability of PVA/BSA to form the "ideal" mixtures were improved:

In dilute saline solution, at pH = 7.4 and temperature of 37 °C, native BSA forms ideal mixtures with PVA (an “inert” macromolecular cosolute [31]), when the water soluble polymer and the globular protein macromolecules exist as individual entities without intermolecular interpenetrating. By increasing the total concentration of macromolecules, in semidilute regime, a phase separation phenomenon occurs and this makes difficult the use of PVA/BSA mixtures for the design of stable composite materials with suitable mechanical properties.

  1. Authors are recommended to perform statistical analysis to prove the significance and reproducibility of the obtained results.

Statistical analysis was presented in the revised manuscript.

  1. Section 4 is missing in this manuscript. I assume that the authors forgot to include a discussion section in the manuscript. A detailed discussion section comparing existing literature and insights must be included in the manuscript.

We corrected the number of different sections. The manuscript was carefully revised and many sentences were reformulated. A discussion comparing existing literature and insights with the present data was included in section 3.

We thank the reviewer for the analysis of the manuscript and for the useful comments and suggestions.

Reviewer 3 Report

Comments and Suggestions for Authors

The submitted manuscript reports the obtention of PVA/BSA-based hydrogels by combining crosslinking procedures and varying the content of PVA as well as adding GP. Despite the authors performing a deep analysis regarding the viscoelastic, rheological, and mechanical properties of polymeric materials, several important aspects related to these kinds of materials and the potential applications proposed need to be addressed. I consider important revisions are needed before acceptance of the article. I am attaching my suggestions and insights below.

1- PVA/BSA should be defined in the article's title.

2- Introduction. Better integration of the concepts related to the different components of the matrices (PVA, GSH, GP) should be made to facilitate readability and understanding as well as to highlight the used strategy of the herein presented materials. 

3- The size and format of the different structures in Scheme 1 should be homogenized.

4- Figure 1. Scale bars are not visible. Please, modify it.

5- The inclusion of hydrogels pictures and also swelling kinetics until equilibrium should be incorporated to compare the different formulations.

6- Also, I suggest the authors perform some measurements demonstrating the hydrophilicity changes between formulations (water contact angle).

7- Experiments related to the degradability of the polymeric matrices are also need to be included in the manuscript. 

8- What about the thermal stability of the hydrogels?

9- All the samples should appear plotted in the different figures.

With kind regards,

Author Response

The submitted manuscript reports the obtention of PVA/BSA-based hydrogels by combining crosslinking procedures and varying the content of PVA as well as adding GP. Despite the authors performing a deep analysis regarding the viscoelastic, rheological, and mechanical properties of polymeric materials, several important aspects related to these kinds of materials and the potential applications proposed need to be addressed. I consider important revisions are needed before acceptance of the article. I am attaching my suggestions and insights below.

1- PVA/BSA should be defined in the article's title.

The title of the manuscript was changed as:

Poly(vinyl alcohol)/ bovine serum albumin hybrid hydrogels with tunable mechanical properties

2- Introduction. Better integration of the concepts related to the different components of the matrices (PVA, GSH, GP) should be made to facilitate readability and understanding as well as to highlight the used strategy of the herein presented materials.

Introduction was revised and improved. Some paragraphs and references were removed.

3- The size and format of the different structures in Scheme 1 should be homogenized.

The different chemical structures from Scheme 1 were revised.

4- Figure 1. Scale bars are not visible. Please, modify it.

We revised Figure 1 and the clarity of SEM immages was improved

5- The inclusion of hydrogels pictures and also swelling kinetics until equilibrium should be incorporated to compare the different formulations.

A new figure with the swelling kinetics for the different formulations was added.

Also, digital pictures of hydrogels were incorporated.

6- Also, I suggest the authors perform some measurements demonstrating the hydrophilicity changes between formulations (water contact angle).

The morphology of the samples has a significant role in terms of contact angle measurement. As we have shown, the investigated hydrogels present a porous structure and thus absorb the liquids very fast. The maximum swelling degree was achieved during the first minutes after immersing the hydrogels in water (between 40 s and 10 min). Consequently, when they were in contact with the drop liquids (water, biological fluids, etc.) used to measure the static contact angles, the drops penetrated into the bulk of the hydrogels, reducing the contact angle to zero degrees. Unfortunately, these measurements are not possible to perform, because the results were hard to compare and tricky to interpret correctly.

7- Experiments related to the degradability of the polymeric matrices are also need to be included in the manuscript.

The hydrogels weight loss after 10 days of imersion in water was evaluated and the results are presented in Table 1.

8- What about the thermal stability of the hydrogels?

The protein addition has no influence on the thermal behavior of PVA hydrogels; however, the physical treatments influence the crystallinity, thermal and mechanical properties [49]. According to our observations and also as it was stated in the literature [23], PVA hydrogels obtained by freezing/thawing method are stable up to approximately 70 °C. Above this temperature, the network structure is compromised due to the fact that PVA starts to dissolve in water.

9- All the samples should appear plotted in the different figures.

The high protein content leads to weak hydrogels which were investigated by rheology in various shear conditions. Supplementary chemical crosslinking was required to ensure the shape fidelity and thus it was possible to investigate the sample in uniaxial compression tests.

We thank the reviewer for the analysis of the manuscript and for the useful comments and suggestions.

Round 2

Reviewer 1 Report

Comments and Suggestions for Authors

This manuscript is good for publication in Polymers

Reviewer 3 Report

Comments and Suggestions for Authors

Thanks to the authors for their responses. The manuscript was considerably improved. I have no more comments to do.

Best,